# Artificial relativistic molecules

Jae Whan Park[1,6], Hyo Sung Kim[1,2,6], Thomas Brumme[3], Thomas Heine[3,4,5] & Han Woong Yeom [1,2]*

We fabricate artificial molecules composed of heavy atom lead on a van der Waals crystal. Pb atoms templated on a honeycomb charge-order superstructure of $IrTe_2$ form clusters ranging from dimers to heptamers including benzene-shaped ring hexamers. Tunneling spectroscopy and electronic structure calculations reveal the formation of unusual relativistic molecular orbitals within the clusters. The spin–orbit coupling is essential both in forming such Dirac electronic states and stabilizing the artificial molecules by reducing the adatom–substrate interaction. Lead atoms are found to be ideally suited for a maximized relativistic effect. This work initiates the use of novel two-dimensional orderings to guide the fabrication of artificial molecules of unprecedented properties.

[1] Center for Artificial Low Dimensional Electronic Systems, Institute for Basic Science (IBS), 77 Cheongam-Ro, Pohang 790-7884, Korea. [2] Department of Physics, Pohang University of Science and Technology, Pohang 790-784, Korea. [3] Department of Chemistry, University of Leipzig, Leipzig, Germany. [4] School of Science, Faculty of Chemistry and Food Chemistry, TU Dresden, 01062 Dresden, Germany. [5] Institute of Resource Ecology, Helmholtz Center Dresden-Rossendorf, Leipzig Research Branch, Permoserstr. 15, 04318 Leipzig, Germany. [6]These authors contributed equally: Jae Whan Park, Hyo Sung Kim. *email: yeom@postech.ac.kr

Artificial atomic clusters or lattices supported on solid surfaces can exhibit tailored electronic, magnetic, and topological properties of fundamental and technological importance. For example, atomic-scale chains and clusters disclosed the quantum confinement of electrons[1–6], topological edge modes[7], superlattice Dirac bands[8], flat bands[9], atomic-scale spin interactions[10], and topological defects[11,12]. The direct atom-by-atom manipulation and the self-assembly of atoms or molecules are two major approaches to realize such atomic-scale chains, clusters, and finite lattices. Both methods, however, have their own limitations under given interatomic interactions, and fabricating energetically and kinetically unfavorable structures or assemblies has been a huge challenge[13].

In case of the self-assembly of supported clusters, such limitations may be overcome by templates, which provide unusual growth environments to produce otherwise unfavored molecular structures. Step edges and one-dimensional surface super-structures were actively used as such templates on metal[14–16] and Si surfaces[17]. The growth of metal atoms on these substrates leads to unusually anisotropic clusters. On the other hand, nanosized two-dimensional superstructures were suggested as novel templates, such as the moiré structure of hexagonal boron nitride[18,19], metal–organic frameworks of DNA[20,21], and metal–organic honeycomb networks on metal surfaces[22–25]. However, while the confinement of metal atoms and molecules on these templates were demonstrated, the formation of artificial molecules was not reported yet.

In this work, we employ a novel template-guided atomic self-assembly technique and take advantage of the strong relativistic effects in lead atoms that effectively create long-ranged interatomic bonds. This allows us to produce a series of unprecedented artificial molecules with relativistic, so called Dirac, electronic orbitals. The honeycomb charge-order superstructure of the van der Waals (vdW) crystal $IrTe_2$ (ref. [26]) cages various Pb clusters ranging from dimers to heptamers, including those in interesting benzene-like hexagonal rings. Atomically resolved spectroscopy and electronic structure calculations reveal clearly the formation of molecular orbitals by direct orbital overlap between neighboring Pb adatoms. The spin–orbit coupling (SOC) of Pb atoms and of the substrate is shown to enhance the interatomic interaction to drive the formation of unprecedentedly relativistic molecules at an unusually large interatomic distance. This case exemplifies the chemical stabilization of nanoscale assemblies uniquely by relativistic effects and opens the potential of novel superstructures on vdW crystals for the fabrication of otherwise unfavored molecules with unprecedented properties.

## Results

**Charge-order superstructure hosts Pb clusters.** Figure 1a shows Pb adatoms deposited on the honeycomb charge-order super-structure at the surface of a $IrTe_2$ vdW crystal. Below 180 K, a honeycomb superstructure forms with a period of 2.2 nm by the spontaneous ordering of $5d^{3+}$ and $5d^{4+}$ valence electrons of Ir (with the buckling of the Te layer accompanied) and competes with various stripe charge orders[26]. Pb adatoms sit at hollow sites surrounded by three Te atoms (Fig. 1b, Supplementary Fig. 2) and, very selectively with yet unclear reasons, inside of each honeycomb, which are above Ir atoms with the $5d^{4+}$ valency (Supplementary Fig. 1). Density-functional theory (DFT) calculations, including SOC consistently predict the four-fold hollow site adsorption with an adsorption energy of $-0.93$ eV against the three-fold hollow site of $-0.74$ eV and Te on-top site of $-0.16$ eV. Clusters ranging from one to seven Pb atoms are confined within a honeycomb unitcell with various lateral configurations. The Pb–Pb interactomic distance is dictated by the

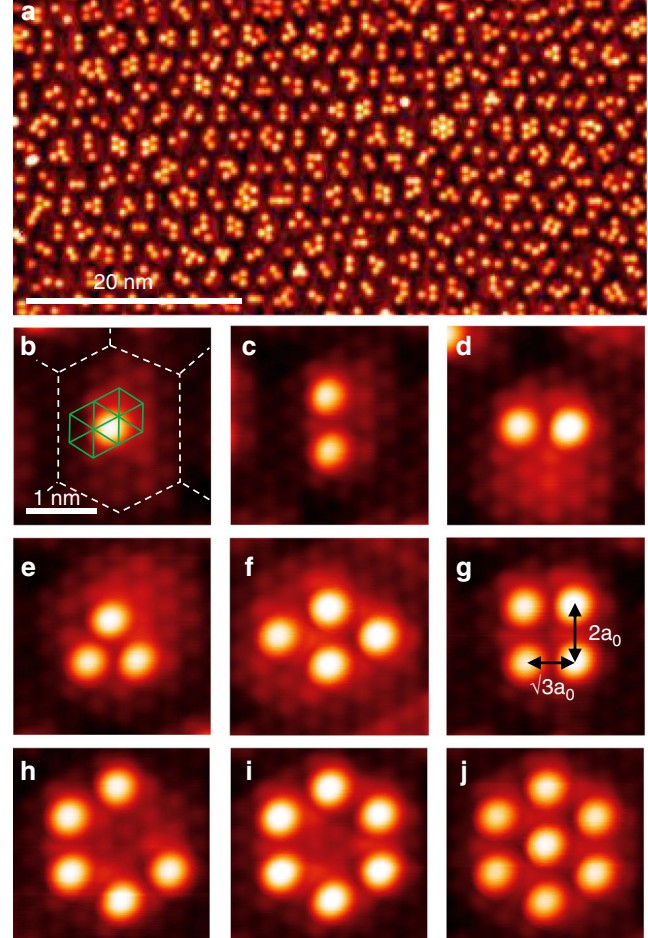

**Fig. 1 STM topography of Pb clusters on $IrTe_2$. a** A typical $IrTe_2$ surface decorated with Pb adatoms in the hexagonal charge order state ($V_s = 1.5$ V, $I_t = 0.05$ nA). Scalebar, 20 nm. **b–j** Various Pb clusters are observed on a single hexagon ($V_s = 20$ mV, $I_t = 1$ nA): **b** monomer (Scalebar, 1 nm), **c** $2a_0$-spaced dimer, **d** $\sqrt{3}a_0$-spaced dimer, **e** $\sqrt{3}a_0$-spaced trimer, **f** $2a_0$-spaced tetramer, **g** mixed-spacing tetramer, **h** $2a_0$-spaced pentamer, **i** $2a_0$-spaced hexamer, and **j** $2a_0$-spaced heptamer. Green solid lines and white dashed lines in **b** denote the Te-(1 × 1) lattice and the charge-order hexagon lattice, respectively.

substrate with two choices of $\sqrt{3}a_0$ or $2a_0$ ($a_0$, the $IrTe_2$ lattice constant of 3.5 Å (ref. [26])). The $2a_0$-spaced dimers (Fig. 1c) are more frequently observed than those of $\sqrt{3}a_0$ Pb–Pb distance (Fig. 1d). As trimers, a heterogeneous trimer with a Pb–Pb spacing both of $\sqrt{3}a_0$ and $2a_0$ or a compact one of $2a_0$ or $\sqrt{3}a_0$ are formed (Fig. 1e, Supplementary Fig. 4c). Figure 1f, g shows a tetramer formed by two $2a_0$ dimers, and one with $\sqrt{3}a_0$ and $2a_0$ dimers, respectively. We also find benzene-like hexagonal ring clusters as shown in Fig. 1h–j among a few other interesting clusters. These configurations are made only by $2a_0$ building blocks and are the largest clusters confined within the hexagonal unit of the template.

**Relativistic molecular orbitals in Pb dimers.** Electronic energy levels of Pb clusters are revealed by scanning tunneling spectroscopy (STS). For a monomer (Fig. 2c), we observe three main spectral features at 1.40, 1.71, and 1.93 eV for Pb $6p$ valence electrons. Our calculation shows that a Pb adatom is partly ionized by donating $6p$ electrons into the substrate to shift the $6p$ states to unoccupied states (Fig. 2o, w). As shown in Fig. 2w, the

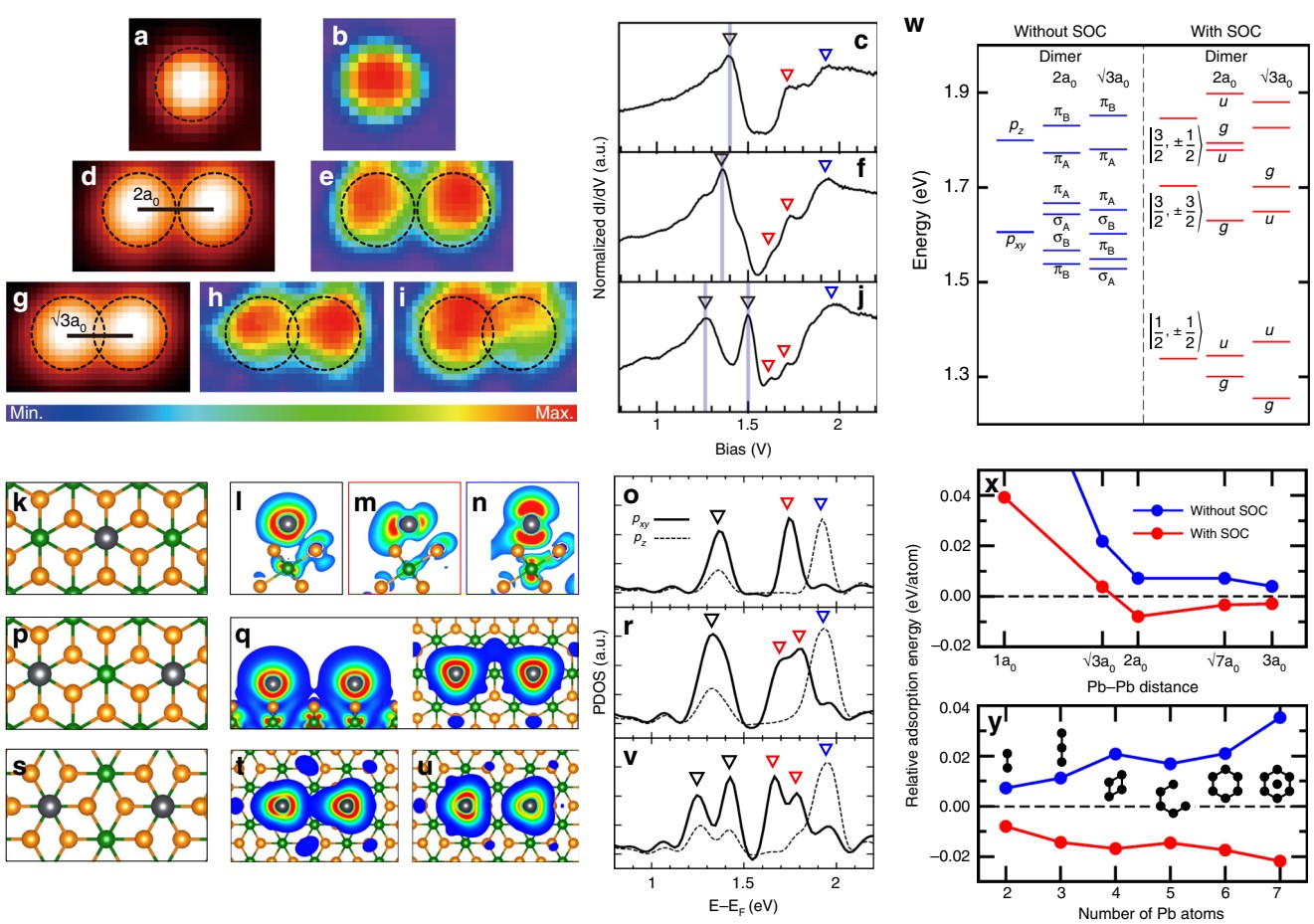

**Fig. 2 Relativistic effects for the Pb monomer and dimers on IrTe₂.** a, d, g STM topography images, b, e, h, i dl/dV maps, and c, f, j STS spectra for a Pb monomer, a $2a_0$-spaced dimer, and a $\sqrt{3}a_0$-spaced dimer, respectively. The dl/dV maps were obtained at the peak positions of STS spectra marked by the inverted black triangles. The corresponding theoretical results are presented in k–v. k, p, s Atomic structures, l–n, q, t, u charge density plots, and o, r, v projected density of states (DOS) for a Pb monomer on the IrTe₂-(5 × 5) supercell, a $2a_0$-spaced dimer, and a $\sqrt{3}a_0$-spaced dimer, respectively. Gray, orange, and green balls in atomic structures represent Pb, Te, and Ir atoms, respectively. Charge densities were obtained for the peak energies of the DOS, three peaks for a monomer, the lowest-energy peak (top and side view) for a $2a_0$-spaced dimer, and the two lowest-energy peaks for a $\sqrt{3}a_0$-spaced dimer. w DFT energy level at Γ point of a Pb monomer and dimers on IrTe₂-(5 × 5) without and with spin–orbit coupling (SOC). For better comparison with experiments, the Fermi level of the Pb/IrTe₂ surface is shifted down by 0.7 eV. Relative adsorption energy per Pb atom of x a Pb dimer on IrTe₂-(7 × 7) as a function of Pb–Pb distance and y various Pb molecules with a Pb–Pb distance of $2a_0$. The adsorption energy of isolated Pb atom set to zero. Blue and red data denote the cases without and with SOC, respectively.

relativistic effect of SOC splits the $6p$ states into $6p_{1/2}$ and $6p_{3/2}$ Dirac orbitals and the latter splits further due to the substrate-induced splitting of in-plane ($p_{xy}$) and out-of-plane ($p_z$) orbitals. Consequently, three main peaks correspond to the relativistic $p$ orbitals of $p_{1/2}$ ($j = 1/2$, $m_j = \pm 1/2$) and $p_{3/2}$ ($j = 3/2$, $m_j = \pm 3/2$, $\pm 1/2$), respectively.

In case of a dimer with a shorter Pb–Pb distance, an energy splitting (0.23 eV) of molecular bonding and antibonding states is observed clearly for the $6p_{1/2}$ state (Fig. 2j). In a longer dimer, the splitting is much smaller, ~0.10 eV, as the orbital overlap reduces (Fig. 2f). Our DFT calculation predicts (Fig. 2r, v) the lowest molecular orbital splittings of 0.05 and 0.18 eV for the longer and shorter dimers, respectively, in reasonable agreement with the experiment. The present DFT calculation seems to underestimate the molecular bonding interaction by ~0.05 eV. These energy scale is consistent with what are expected for a freestanding Pb–Pb dimer (Supplementary Fig. 6). This result clearly indicates the molecular orbital formation within the Pb clusters at an unusually large Pb–Pb distance of 7 Å, which is observed consistently for all cluster configurations identified (Supplementary Fig. 7).

The molecular levels of a Pb dimer are well explained by the simple tight-binding interaction of Dirac atomic orbitals[27] perturbed weakly by the substrate as discussed above. That is, three relativistic $p$ orbitals split simply into the combination of molecular orbitals in gerade ($g$) and ungerade ($u$) symmetries (Fig. 2w). For example, the two lowest peaks correspond to the combination of the $p_{1/2}$ orbitals in $g$ and $u$ symmetries. The resulting Dirac molecular orbitals are clearly differentiated from their scalar-relativistic counterparts of $\sigma$ and $\pi$ molecular orbitals. The $g$ ($u$) orbital consists of one-third $\sigma$-bond ($\sigma$-antibond) and two-third $\pi$-antibond ($\pi$-bond) character, indicating a substantial mixing between $\sigma$- and $\pi$-bonds due to the full relativity effect of SOC. The Dirac basis and the interaction with substrate atoms also breaks the mirror symmetry perpendicular to the dimer axis. This is reasonably confirmed in the experiment (Fig. 2h, i) and in the calculation (Fig. 2t, u). Further details of the interaction with the substrate will be discussed below, which become important for the understanding of the relativistic effect on the adsorption energy.

The direct and real space observation of relativistic molecular orbitals is exceptional, since most of previous experiments for

very few heavy element molecules relied on spectroscopy techniques[28–31] and, thus, the relativistic effects were mainly discussed in theoretical aspects[32,33]. Although a Pb dimer has a much longer bond length (7 Å) than other relativistic molecules[33–35] and there is a substantial adatom–substrate interaction, the molecular splitting mainly arises from the direct overlap of relativistic $p$ orbitals. This is distinguished from the substrate-mediated long-range interaction of other metal clusters on metal surfaces[36,37].

**Mechanism of the interaction between Pb atoms**. It is important to point out that the attractive interaction between Pb adatoms is only due to SOC and that the $2a_0$ distance imposed by the substrate is optimal for the formation of a Pb dimer (Fig. 2x) in our calculations. This is consistent with the observation of the higher population of $2a_0$ dimers. The SOC energetics for the larger Pb molecules, from trimer to heptamer (Fig. 2y, Supplementary Fig. 10), is also consistent with the experimental observation of larger Pb molecules with exclusively $2a_0$ nearest-neighbor distance. The $2a_0$ distance is optimized by two competing energy contributions, the ionic (or dipole) repulsion and the orbital overlap energy gain of adatom–substrate hybridized states. The comparison of the calculated differential charges with and without SOC reveals two sophisticated SOC effects for those contributions; (i) a reduced donation of $p$ electrons to the substrate to reduce the ionic repulsion and (ii) an enhanced orbital overlap of the Pb–Te hybridized states (Supplementary Fig. 10). The former is due to the lowering of the atomic energy level of Dirac orbitals by SOC. It is also partly due to the charge redistribution of the IrTe$_2$ substrate by SOC, namely the charge transfer from the $d_{z^2}$ states of Ir atom to Te atoms. The substantial interaction of Pb with the substrate Te orbitals provides an extra energy gain for the antibonding level of the $p_{1/2}$ ($m_j = \pm 3/2$) orbital of the $\sqrt{3}a_0$ dimer (Supplementary Fig. 9b). That is, the interwined effect of the substrate and the relativity determines the stability of the Pb molecules. In these respects, Pb is ideal since it has very strong relativistic effects (in comparison, for example, to Sn, Supplementary Fig. 11) and a proper interaction strength with the substrate (in comparison to Tl, Supplementary Fig. 12). Of course, the formation of Pb molecules would also be affected by the size and shape of the charge-order honeycomb template.

**Benzene-like ring-shaped Dirac molecules**. While a consistent interaction mechanism holds for all 15 different Pb molecules identified (Supplementary Fig. 4), particularly interesting ones are those based on a six-fold hexagonal ring (Fig. 3). The benzene-like ring molecule of Pb$_6$ with a $2a_0$ Pb–Pb distance exhibit the weak molecular splitting similar to a $2a_0$ dimer (Fig. 3e). The intriguing molecular orbital formation is clearly imaged by the $dI/dV$ maps: (i) the broken AB sublattice symmetry for the lowest-energy state (Fig. 3b), (ii) a Kekulé-like distortion for the second state (Fig. 3c), and (iii) a mirror symmetry-broken two-fold-symmetric bonding feature for the third state (Fig. 3d). Our calculation reproduces well the spatial characteristics of these molecular orbitals, which are indeed due to the relativistic character and the interaction with the substrate (Fig. 3l–n, Supplementary Fig. 13). Note, however, the remaining energetic discrepancy between experiment and calculation. The present calculations underestimate the molecular bonding interaction and the calculated spectral features are almost degenerate at ~1.3 eV for the benzene-like molecule. We think that this difference is at least partly due to the limitation of our model in taking into account of the substrate electronic states in its charge-ordered correlated state[26]. Another interesting molecule is the Pb pentamer (Fig. 3f–j), whose Dirac molecular orbitals are also

reasonably well simulated (Fig. 3p–t). Such type of an individual open ring molecule cannot be synthesized. The Pb$_5$ molecule features the edge states at both truncated ends between the bonding and antibonding levels of three Dirac $p$ orbitals (Supplementary Fig. 14). It indicates the ring-type Pb molecules are circularly delocalized electron systems. The formation of a filled-benzene-ring molecule is also highly unusual (Fig. 1j), where the interatomic overlap is much enhanced by the central Pb atom with unique bonding configurations (Supplementary Figs. 15–17). This molecule can be compared with a rare example of a planar B$_7$ molecule[38] except for the strong SOC. While not accessible by the present experiments, the benzene-type Dirac molecular orbitals fabricated here have unique and interesting spin configurations (Supplementary Fig. 13), which may be exploited further. It becomes very obvious that the present artificial Pb molecules introduce unprecedented molecular configurations combined with the strong SOC.

**Summary**. We fabricate various artificial Pb molecules utilizing a novel honeycomb template of the charge-ordered superstructure in IrTe$_2$. The molecular orbital formation is clearly observed in the Pb clusters from dimers to unusual benzene-ring-type molecules. The SOC is found to play crucial roles in both the condensation of Pb atoms and the molecular orbital formation, bringing them into an unprecedented regime of relativistic Dirac molecules. Beyond the relativistic chemistry, the unusual bonding and spin structure of these relativistic molecular orbitals may be combined with the novel electronic properties of the substrate, such as the charge ordering and the emerging superconductivity[26] to lead to a new type of a quantum system. In addition to the versatile technique of atom-by-atom manipulation, where the interatomic distance may be varied, the use of various types of 2D superstructures as templates would definitely extend the potential of molecular or cluster self assemblies. Interesting 2D superstructures can include twisted bilayer graphene[39] and domain-wall or twin-boundary networks of transition metal dichalcogenides[40,41]. The templated self-assembly of magnetic atoms on such substrates can be very interesting due to the proximity to the novel 2D electronic states of substrates.

## Methods

**Sample preparation**. Single crystals of IrTe$_2$ were grown by Te flux using pre-sintered IrTe$_2$ polycrystals, as reported previously. Samples were cleaved in a vacuum better than $5 \times 10^{-10}$ torr at room temperature. Pb atoms are grown on cleaved IrTe$_2$ surfaces at room temperature. The Pb atom tends to cover the IrTe$_2$ surface up to 0.1 mL. Over this coverage, this Pb growth mode depends on the deposition temperature. For example, when the Pb grows at room temperature, Pb forms a uniform film. However, at low-temperature deposition, depending on the coverage and annealing condition, we can found different heights of the Pb islands.

**STM measurement**. All the STM measurements were performed with a commercial ultrahigh vacuum cryogenic STM (Specs, Germany) in the constant-current mode with PtIr tips at 4.3 K. The differential conductance, $dI/dV$, was measured using the lock-in detection with a modulation of 1.17 kHz.

**DFT calculations**. We perform the relativistic DFT calculations using the Vienna abinitio simulation package[42] within the generalized gradient approximation of the Perdew–Burke–Ernerhof type[43] and the projector augmented-wave method[44]. The IrTe$_2$ surface is modeled by a $(5 \times 5)$ and $(7 \times 7)$ supercell with a single IrTe$_2$ layer and a vacuum spacing of ~23.6 Å. The calculated value 3.839 Å is used as the lattice constant of IrTe$_2$. We use a plane-wave basis with a cutoff of 211 eV for expansion of electronic wave functions and a $15 \times 15 \times 1$ $k$-point mesh for the $1 \times 1$ Brillouin-zone integrations. To avoid the unexpected substrate distortion of the IrTe$_2$, only adsorbed Pb atoms are relaxed until the residual force components are within 0.02 eV/Å. Here, the Fermi level was shifted down by 0.70 eV because to overcome the inherent discrepancy between experiment and theory due to unknown charge-ordered structure, doping effect of the Pb adsorbates and the IrTe$_2$ sample and/or the inaccuracy of DFT band-energy calculations.

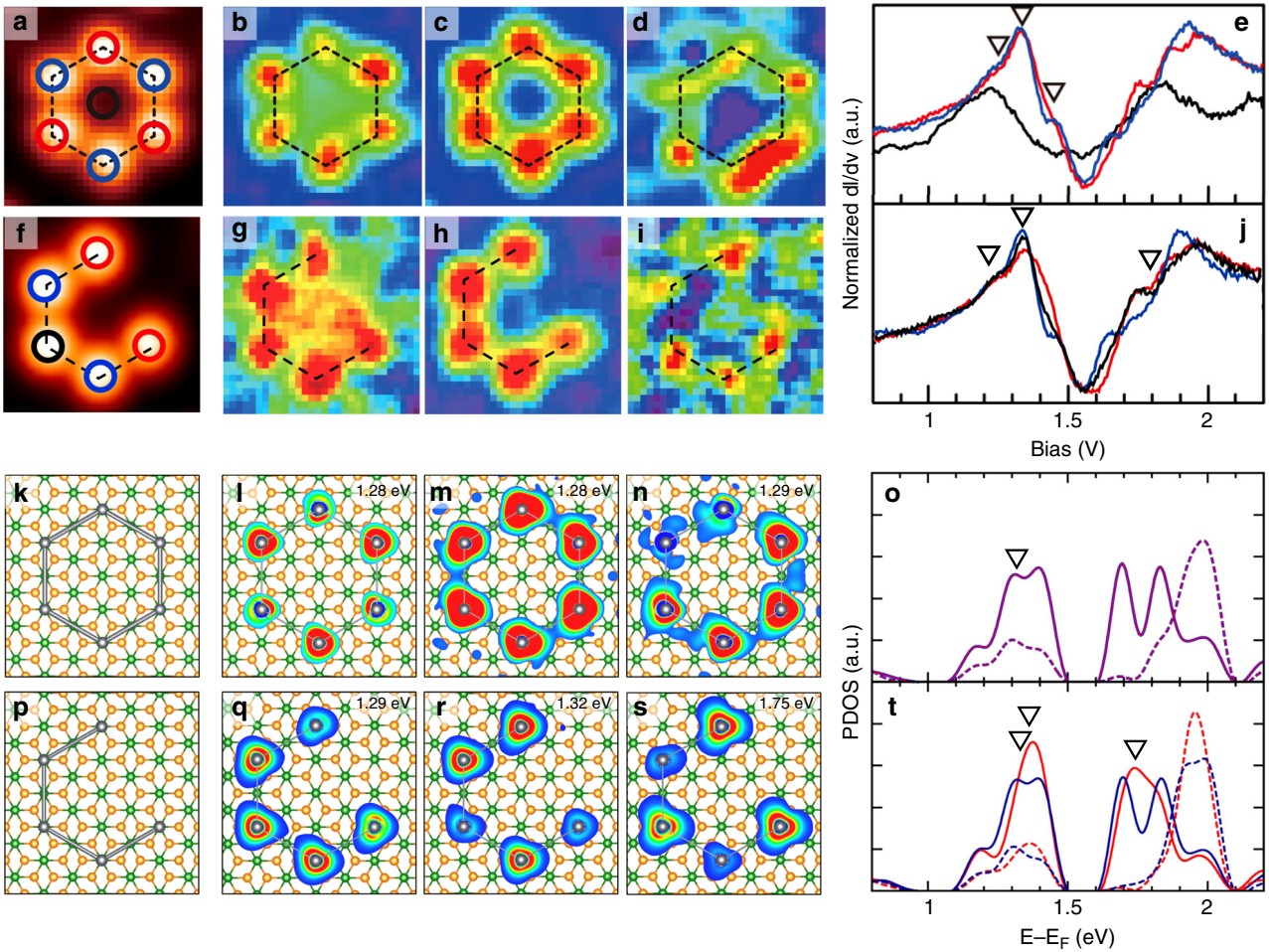

**Fig. 3 Benzene-ring-shaped Pb clusters on IrTe₂. a, f** STM topography, **b–d, g–i** dI/dV maps, and **e, j** STS spectra for a Pb hexamer and a pentamer, respectively. dI/dV maps are taken at the energies indicated in the STS spectra. Red, blue, and black lines in the STS spectra are obtained at the red, blue, and black Pb atoms indicated in **a** and **f**. The corresponding theoretical results are presented in **k–t. k, p** Atomic structure, **l–n, q–s** charge densities at the given energies (spin decoupled and spin summed, respectively), and **o, t** projected DOS, for a Pb hexamer and a pentamer on IrTe₂-(7 × 7), respectively. For the Pb hexamer, the localized DOS at red and blue Pb atoms is identical to each other. For the Pb pentamer, the red (dark blue) line denote the localized states at truncated end atoms (the other Pb atoms). Solid and dashed lines represent $p_{xy}$ and $p_z$ states, respectively.

## Data availability

The authors declare that the data supporting the findings of this study are available within the article and its Supplementary Information.

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

## Acknowledgements

J.W.P., H.S.K., and H.W.Y. were supported by Institute for Basic Science (grant no. IBS-R014-D1). The IrTe$_2$ crystal was provided by S.W. Cheong in Rutgers University. J. Lee is appreciated for his help in data processing. T.B. and T.H. used the ZIH Dresden computational resources.

## Author contributions

J.W.P. performed the DFT calculations. H.S.K. carried out the STM measurements. J.W.P. and H.W.Y. analyzed the data, and wrote the manuscript with the comments of all other authors. H.W.Y. supervised the research.

## Competing interests

The authors declare no competing interests.
