## [Peer Review File · Nature Communications]

Reviewers' comments:

Reviewer #1 (Remarks to the Author):

The ms. by Jae Whan Park et al. describes the formation and characterization of several different structures consisting of Pb atoms on the IrTe₂ surface. These dimers, trimers etc. (up to heptamers) behave as artificial atoms with a level structure that can be understood in terms of relativistic molecular orbitals. The manuscript describes exciting experiments and interesting, novel results. However, I disagree with the angle taken in the introduction and do not agree with some of the claims of novelty. If this can be resolved, the paper could be accepted in Nature Communications.

The problem I have with the introduction starts with the following statement: "...interatomic interactions, which is largely not manipulatable but determined by fundamental rules of physics" – This is obviously correct, but I do not really understand the point the authors are trying to make. On the IrTe₂ surface, the Pb adatoms are occupying the hollow sites and the structures that are formed by self-assembly consist of various non-close packed structures. Such structures could certainly be reproduced by STM-based atom manipulation. While manipulation on IrTe₂ surface may be difficult, Pb atom manipulation on, for example, Cu has been described (e.g. G. Meyer et al. Applied Physics A 68, 125 (1999)). In addition to the assembly, IrTe₂ might naturally serve other roles as well, e.g. be important for decoupling the Pb atom states from the substrate (see questions below).

The introduction of this manuscript somehow claims to go beyond atom manipulation or self-assembly in making atomic scale structures – I disagree with this. I think this is a simple example of templated assembly, which in other forms has been used to "make" artificial molecules also in the past: Fullerene (Nature 424, 1029 (2003), Nat. Mater. 3, 229 (2004)), Xe atoms (Nat. Commun. 6, 6071 (2015), Small 12, 3757 (2016)), CoPc molecules (J. Phys. Chem. 120, 8772 (2016)), porphyrin and TCNQ (ACS Nano 8, 430 (2018)) and so on. Hence, I don't find the formation of these loosely-packed Pb clusters surprising. The reason for the site-selective templating on IrTe₂ surface might be, for example, work function modulation (that creates an in-plane electric field) that surely exists on such surface. The work function modulation on the clean IrTe₂ surface could be probed by e.g. field-emission resonances.

Having said all that, I do find the study of the artificial molecules with heavy spin-orbit interaction interesting and worthwhile publication in Nature Communications. Here I have a couple of minor comments/questions as follows:

1. Is there a more intuitive way of understanding the molecular orbitals formed from the atomic states of the Pb atoms? Is it possible to understand the artificial molecular states in a tight-binding picture? While DFT calculations are nice, they do not really give intuitive understanding of the formed states.
2. The authors should consider further experiments on some other Te- (or Se-) terminated surface where the lattice constant is different from IrTe₂. This would allow tuning the distance between the Pb atoms and consequently, the strength of the hybridization between them.
3. What is the role of IrTe₂ in the electronic coupling between the Pb atoms? Is there any coupling to the IrTe₂ states?
4. There is disorder in the LDOS maps in Figs. 2 G-I and 3A-I – where is it coming from? From the substrate (it seems that the hexagonal pattern in Fig. S1 C is not completely regular)? Do the atom-to-atom distance vary? Would this have an effect in the simulated response?
5. Schematics in Fig. S1 are unclear and should be improved.

Reviewers' comments:

Reviewer #1 (Remarks to the Author):

The ms. by Jae Whan Park et al. describes the formation and characterization of several different structures consisting of Pb atoms on the IrTe₂ surface. These dimers, trimers etc. (up to heptamers) behave as artificial atoms with a level structure that can be understood in terms of relativistic molecular orbitals. The manuscript describes exciting experiments and interesting, novel results. However, I disagree with the angle taken in the introduction and do not agree with some of the claims of novelty. If this can be resolved, the paper could be accepted in Nature Communications.

The problem I have with the introduction starts with the following statement: "...interatomic interactions, which is largely not manipulatable but determined by fundamental rules of physics" – This is obviously correct, but I do not really understand the point the authors are trying to make. On the IrTe₂ surface, the Pb adatoms are occupying the hollow sites and the structures that are formed by self-assembly consist of various non-close packed structures. Such structures could certainly be reproduced by STM-based atom manipulation. While manipulation on IrTe₂ surface may be difficult, Pb atom manipulation on, for example, Cu has been described (e.g. G. Meyer et al. Applied Physics A 68, 125 (1999)). In addition to the assembly, IrTe₂ might naturally serve other roles as well, e.g. be important for decoupling the Pb atom states from the substrate (see questions below).

The introduction of this manuscript somehow claims to go beyond atom manipulation or self-assembly in making atomic scale structures – I disagree with this. I think this is a simple example of templated assembly, which in other forms has been used to "make" artificial molecules also in the past: Fullerene (Nature 424, 1029 (2003), Nat. Mater. 3, 229 (2004)), Xe atoms (Nat. Commun. 6, 6071 (2015), Small 12, 3757 (2016)), CoPc molecules (J. Phys. Chem. 120, 8772 (2016)), porphyrin and TCNQ (ACS Nano 8, 430 (2018)) and so on. Hence, I don't find the formation of these loosely-packed Pb clusters surprising. The reason for the site-selective templating on IrTe₂ surface might be, for example, work function modulation (that creates an in-plane electric field) that surely exists on such surface. The work function modulation on the clean IrTe₂ surface could be probed by e.g. field-emission resonances.

Having said all that, I do find the study of the artificial molecules with heavy spin-orbit interaction interesting and worthwhile publication in Nature Communications. Here I have a couple of minor comments/questions as follows:

1. Is there a more intuitive way of understanding the molecular orbitals formed from the atomic states of the Pb atoms? Is it possible to understand the artificial molecular states in a tight-binding picture? While DFT calculations are nice, they do not really give intuitive understanding of the formed states.
2. The authors should consider further experiments on some other Te- (or Se-) terminated surface where the lattice constant is different from IrTe₂. This would allow tuning the distance between the Pb atoms and consequently, the strength of the hybridization between them.
3. What is the role of IrTe₂ in the electronic coupling between the Pb atoms? Is there any coupling to the IrTe₂ states?
4. There is disorder in the LDOS maps in Figs. 2 G-I and 3A-I – where is it coming from? From the substrate (it seems that the hexagonal pattern in Fig. S1 C is not completely regular)? Do the atom-to-atom distance vary? Would this have an effect in the simulated response?
5. Schematics in Fig. S1 are unclear and should be improved.

Reviewer #2 (Remarks to the Author):

The article of Park et al. reports an interesting study of Pb clusters assembled on the charge density wave material IrTe₂.

I read the manuscript with great interest. I think it is a very solid study comparing high quality STM/STS data with state-of-the-art ab-initio calculation. The authors show the importance of the relativistic spin-orbit interaction in explaining the electronic properties of the clusters.

I have not any comment on the scientific aspects of the manuscript. It is clearly written, and I would not have any objection in publishing it in its current state.

However, I find it suitable for a more specialized journal. I do not see sufficient potential for impact and novelty as expected by Nature Communication. I would be happy to change my opinion if these clusters could be controllably manipulated by the STM tip, to achieve a real tailored-made fabrication approach. At present, the clusters are randomly obtained by dosing Pb on IrTe₂. Alternatively, another interesting aspect would be the creation of a well-defined artificial Pb superlattice.

I also find the title a little bit confusing. The article highlights the importance of the spin-orbit coupling. Since Pb is an heavy element, I do not find this aspect very surprising. However, I think that naming these clusters "Dirac molecules" can be rather misleading. It recalls a concept generally used in other contexts (Dirac bands in graphene or topological materials).

To summarize: I think this is a very comprehensive study, but I do not think it fits within the scope of the journal. However, if the editor or the other referees disagree on this aspect, I have no objection in publishing the manuscript in its present state.

Reviewer #3 (Remarks to the Author):

The authors reported a comprehensive study on the interactions among individual Pb adatoms on IrTe₂ using STM and DFT. The authors concluded that small clusters of Pb in several configurations form molecular orbitals supported by STS spectra, real space STS mapping, and DFT calculations. The results are interesting and likely correct except that the energies of the simulated charge density maps which are shown to resemble experiments do not quite agree with experiments. Still, this study provides a simple platform for exploring artificial states of matter via bottom-up methods. Compared to another bottom-up system of "molecular graphene" from manipulated CO molecules (Nature 483, 306–310, 2012), the drawback in the current work is clearly the lack of flexibility in the inter-atomic distances and lattice structures. For example, could the inter-Pb distance be manipulated by STM tip in this case? I also have the following questions the authors should address before this manuscript is considered for publication.

1. The authors claimed, "The direct observations of relativistic molecular orbitals are exceptional". While the DFT calculations provided an explanation for the experimental data, in the strictest sense, the observation should be stated as consistent with relativistic molecular orbitals.

2. In Fig. 3B, the authors claimed broken A-B sublattice symmetry. I don't see such evidence clearly from the map in Fig. 3B.

3. Does Fig. 3L-N correspond to panels B-D, respectively? Why are Fig. 3L and M different while they are both simulated at 1.28eV as the authors labeled? It looks to me that Fig. 3B-D are taken at very different energies (the black arrows in E), but Fig. 3L-N are given at energies of 1.28 and 1.29 eV, which are close. Though L-N resembles B-D, the large energy difference makes it not convincing.

4. Similarly, Fig. S16 B-D seems to have very different energies than what are indicated in Fig. S16 G-I.

5. Explanations are needed for Fig. 3O, T as they are not mentioned in the main texts.

6. In the caption of Fig. S13, the authors mentioned spin configurations in Fig.4, while there is no Fig. 4 in this manuscript. The authors also mentioned "unusual spin structure" in the Summary section, which is not discussed at all in the manuscript.

7. More explanations of Fig. S13 are needed. For example, what are the energies of the simulated charge density distributions? Does charge distributions with SOC reproduce experimental

observations better than without SOC?

8. In terms of materials systems, the authors are encouraged to provide outlooks on what other adsorbates and substrates are interesting beyond what's demonstrated here.

Reviewer #2 (Remarks to the Author):

The article of Park et al. reports an interesting study of Pb clusters assembled on the charge density wave material IrTe₂.

I read the manuscript with great interest. I think it is a very solid study comparing high quality STM/STS data with state-of-the-art ab-initio calculation. The authors show the importance of the relativistic spin-orbit interaction in explaining the electronic properties of the clusters.

I have not any comment on the scientific aspects of the manuscript. It is clearly written, and I would not have any objection in publishing it in its current state.

However, I find it suitable for a more specialized journal. I do not see sufficient potential for impact and novelty as expected by Nature Communication. I would be happy to change my opinion if these clusters could be controllably manipulated by the STM tip, to achieve a real tailored-made fabrication approach. At present, the clusters are randomly obtained by dosing Pb on IrTe₂. Alternatively, another interesting aspect would be the creation of a well-defined artificial Pb superlattice.

I also find the title a little bit confusing. The article highlights the importance of the spin-orbit coupling. Since Pb is an heavy element, I do not find this aspect very surprising. However, I think that naming these clusters "Dirac molecules" can be rather misleading. It recalls a concept generally used in other contexts (Dirac bands in graphene or topological materials).

To summarize: I think this is a very comprehensive study, but I do not think it fits within the scope of the journal. However, if the editor or the other referees disagree on this aspect, I have no objection in publishing the manuscript in its present state.

Reviewer #3 (Remarks to the Author):

The authors reported a comprehensive study on the interactions among individual Pb adatoms on IrTe₂ using STM and DFT. The authors concluded that small clusters of Pb in several configurations form molecular orbitals supported by STS spectra, real space STS mapping, and DFT calculations. The results are interesting and likely correct except that the energies of the simulated charge density maps which are shown to resemble experiments do not quite agree with experiments. Still, this study provides a simple platform for exploring artificial states of matter via bottom-up methods. Compared to another bottom-up system of "molecular graphene" from manipulated CO molecules (Nature 483, 306–310, 2012), the drawback in the current work is clearly the lack of flexibility in the inter-atomic distances and lattice structures. For example, could the inter-Pb distance be manipulated by STM tip in this case? I also have the following questions the authors should address before this manuscript is considered for publication.

1. The authors claimed, "The direct observations of relativistic molecular orbitals are exceptional". While the DFT calculations provided an explanation for the experimental data, in the strictest sense, the observation should be stated as consistent with relativistic molecular orbitals.

2. In Fig. 3B, the authors claimed broken A-B sublattice symmetry. I don't see such evidence clearly from the map in Fig. 3B.

3. Does Fig. 3L-N correspond to panels B-D, respectively? Why are Fig. 3L and M different while they are both simulated at 1.28eV as the authors labeled? It looks to me that Fig. 3B-D are taken at very different energies (the black arrows in E), but Fig. 3L-N are given at energies of 1.28 and 1.29 eV, which are close. Though L-N resembles B-D, the large energy difference makes it not convincing.

4. Similarly, Fig. S16 B-D seems to have very different energies than what are indicated in Fig. S16 G-I.

5. Explanations are needed for Fig. 3O, T as they are not mentioned in the main texts.
6. In the caption of Fig. S13, the authors mentioned spin configurations in Fig.4, while there is no Fig. 4 in this manuscript. The authors also mentioned "unusual spin structure" in the Summary section, which is not discussed at all in the manuscript.
7. More explanations of Fig. S13 are needed. For example, what are the energies of the simulated charge density distributions? Does charge distributions with SOC reproduce experimental observations better than without SOC?
8. In terms of materials systems, the authors are encouraged to provide outlooks on what other adsorbates and substrates are interesting beyond what's demonstrated here.

Replies to the comments of the first referee

The ms. by Jae Whan Park et al. describes the formation and characterization of several different structures consisting of Pb atoms on the IrTe₂ surface. These dimers, trimers etc. (up to heptamers) behave as artificial atoms with a level structure that can be understood in terms of relativistic molecular orbitals. The manuscript describes exciting experiments and interesting, novel results. However, I disagree with the angle taken in the introduction and do not agree with some of the claims of novelty. If this can be resolved, the paper could be accepted in Nature Communications.

The problem I have with the introduction starts with the following statement: "...interatomic interactions, which is largely not manipulatable but determined by fundamental rules of physics" – This is obviously correct, but I do not really understand the point the authors are trying to make. On the IrTe₂ surface, the Pb adatoms are occupying the hollow sites and the structures that are formed by self-assembly consist of various non-close packed structures. Such structures could certainly be reproduced by STM-based atom manipulation. While manipulation on IrTe₂ surface may be difficult, Pb atom manipulation on, for example, Cu has been described (e.g. G. Meyer et al. Applied Physics A 68, 125 (1999)). In addition to the assembly, IrTe₂ might naturally serve other roles as well, e.g. be important for decoupling the Pb atom states from the substrate (see questions below).

The introduction of this manuscript somehow claims to go beyond atom manipulation or self-assembly in making atomic scale structures – I disagree with this. I think this is a simple example of templated assembly, which in other forms has been used to "make" artificial molecules also in the past: Fullerene (Nature 424, 1029 (2003), Nat. Mater. 3, 229 (2004)), Xe atoms (Nat. Commun. 6, 6071 (2015), Small 12, 3757 (2016)), CoPc molecules (J. Phys. Chem. 120, 8772 (2016)), porphyrin and TCNQ (ACS Nano 8, 430 (2018)) and so on. Hence, I don't find the formation of these loosely-packed Pb clusters surprising. The reason for the site-selective templating on IrTe₂ surface might be, for example, work function modulation (that creates an in-plane electric field) that surely exists on such surface. The work function modulation on the clean IrTe₂ surface could be probed by e.g. field-emission resonances.

We largely agree on the point of the referee about our claim of the interatomic interaction. Unintentionally, our previous writing was misleading in claiming that our method goes generally

beyond the self assembly and the atom manipulation by controlling the interatomic interaction. Now the revised introduction focuses on our particular achievement, extending the templated self-assembly by making unprecedented relativistic molecular bonding structures within a new type of a 2D template.

Having said all that, I do find the study of the artificial molecules with heavy spin-orbit interaction interesting and worthwhile publication in Nature Communications. Here I have a couple of minor comments/questions as follows:

1. Is there a more intuitive way of understanding the molecular orbitals formed from the atomic states of the Pb atoms? Is it possible to understand the artificial molecular states in a tight-binding picture? While DFT calculations are nice, they do not really give intuitive understanding of the formed states.

We agree generally that DFT might not give straightforward intuitive understanding. This is partly true for the present case where the interaction with the substrate is fairly intriguing. However, except for the charge transfer between Pb and IrTe₂, which pushes up the $|1/2, \pm 1/2\rangle$ in energy, the orbital basis of 6p electrons for a single Pb adatom on the substrate is simply established as the relativistic (Dirac) atomic orbitals of $|1/2, \pm 1/2\rangle$, $|3/2, \pm 3/2\rangle$ and $|3/2, \pm 1/2\rangle$ (through the interaction with the substrate, which requires DFT calculation, see Supplementary Fig. 9B and our reply 3 below). Then, the molecular bonding can be easily described within the tight binding picture (gerade and ungerade molecular orbital formation) as given in Fig. 2w due to the relatively weak Pb-Pb interaction. That is, the tight binding picture is, in essence, already given in Fig. 2w. This point is extensively explained below as our reply 3. In order to deliver this point, the simple bonding picture, more clearly, we refined the corresponding explanation in the main text and revised the corresponding figure in the supplements.

2. The authors should consider further experiments on some other Te- (or Se-) terminated surface where the lattice constant is different from IrTe₂. This would allow tuning the distance between the Pb atoms and consequently, the strength of the hybridization between them.

We appreciate this constructive suggestion. However, note that the Pb-Pb bond length difference tunes only the tight binding molecular level splitting (as explained above and below) but not the fundamental relativistic nature of the bonding. The change of the substrate, on the other hand, would change the Pb-substrate interaction, thus the relativistic orbital basis can change, deviating substantially from the present case. This is because the simple orbital basis of the present case (being very close to the atomic one) comes from the fact that the Pb 6p electrons are located accidentally within the energy gap of the

IrTe₂ substrate (see figure below).

LDOS of IrTe₂, Pb/IrTe₂ and TI/IrTe₂.

Moreover, the charge order honeycomb template is usually not available in other substrate and thus the cluster formation is not expected to happen. In many case, Pb atoms aggregate into compact islands as we have tested on Pb/graphene, Pb/Bi₂Se₃ and Pb/Fe₃Ge₂Te₅. Therefore, we believe that there is little reason to compare the present case with Pb adsorbates on other substrates.

3. What is the role if IrTe₂ in the electronic coupling between the Pb atoms? Is there any coupling to the IrTe₂ states?

We appreciate this important comment, which is related to the most important part of the present work as already exposed above. We regret that this point was not sufficiently clear in our original manuscript. Basically, the Pb-Pb bondings on the IrTe₂ substrate can be explained by the direct molecular orbital formation of weakly perturbed Pb atoms. We provide a detailed explanation of the Pb-substrate interaction below and the manuscript was revised to present this more clearly.

The interaction of Pb with the IrTe₂ substrate to form relativistic atomic orbitals:

1. The 6p electrons of Pb atom transfer into the substrate, which shift the $|1/2\rangle$ states to the empty state close to $|3/2\rangle$ states. (Stated in the main text)
2. The $|3/2\rangle$ states naturally split into in-plane ($|3/2, \pm 3/2\rangle$) and out-of-plane ($|3/2, \pm 1/2\rangle$) orbitals due to the substrate. (Stated in the main text)
3. The interaction with the substrate makes the bonding-antibonding (g-u) splitting larger than isolated

Pb molecules. For example, the $|1/2\rangle$ relativistic state is composed of one-third σ bond and two-third π antibond. The substrate saturates the out-of-plane orbitals and thus the contribution of the π antibond part to the $|1/2\rangle$ state is reduced by the substrate. The reduction of the antibonding part enhances the total bonding energy. (Stated in the main text. See the figure below.)

Therefore, the Pb-Pb bondings on the IrTe₂ substrate can be explained by the direct molecular orbital formation of weakly perturbed Pb atoms.

PDOS of the Pb monomer and dimers on Pb/IrTe₂ and without substrate

(Only in J, K, and L, the $|1/2\rangle$ state appears within the energy range due to the charge transfer to the substrate.)

Charge characters (u and g) of the Pb($\sqrt{3}$ dimer)/IrTe₂ in Supplementary Fig. 9

4. There is disorder in the LDOS maps in Figs. 2 G-I and 3A-I – where is it coming from? From the substrate (it seems that the hexagonal pattern in Fig. S1 C is not completely regular)? Do the atom-to-atom distance vary? Would this have an effect in the simulated response?

Usually, the LDOS maps in STM is much more noisy than the topographic image since it relies on weak

differential tunneling conductance signal and thus the lock-in-technique is essential. Namely, the noise is intrinsic to the measurements. A much longer data taking could reduce the noise but it requires about ten times longer measurements for noticeable improvement. As one can find below, the noise (more precisely, the irregularity of the LDOS maps) is still manageable to draw major qualitative conclusions through the comparison with the corresponding simulations (see figure below).

Fig. 2h and 2i. Purple dashed (strong) and solid (weak) lines are guides for eye.

The situation is little bit more complicated in Fig. 3 for the larger clusters, where a large number of molecular levels with small splittings coexist within a small energy window (see figure below). The intermixing between neighboring energy levels is unavoidable within the experimental resolution and can cause the rather ill defined shapes in LDOS maps in comparison with the simulated results. This point is explicitly commented in the revised manuscript for reader's understanding.

STS spectra and PDOS of the Pb hexamer

On the other hand, as the referee pointed out, the hexagonal template is not regular, which may affect the number and the configuration of Pb atoms in each hexagon. However, Pb-Pb distance is quite regular as it is governed by the 1x1 lattice of the substrate (as far as judged from the STM topography data, see the figure below). There may be certain possibility for the irregular hexagonal pattern to affect the electronic states of large clusters, which may be related to the referee's concern. This possibility, however, was not noticed in our experimental data sets.

Fig. 1i. Regular hexagon of Pb₆ molecule

5. Schematics in Fig. S1 are unclear and should be improved.

We thank the referee for this helpful comment. According to the suggestion, we revised Supplementary Fig. 1 with better schematics of the atomic structures as shown below.

Supplementary Fig. 1

SFIG. 1: Site selective Pb adsorption on the dimerized IrTe₂ substrate. **a** STM image ($V_s = 5$ mV, $I_t = 1$ nA) and **b** dimerized structural model of the stripe phase. **c** STM image ($V_s = 10$ mV, $I_t = 1$ nA) and **d** dimerized structural model of the honeycomb charge order phase (top and side views). The STM topographies show the charge ordering induced by the dimerization of the Ir atoms. The dimerized Ir atoms (red circles) has $5d^{4+}$ valence electrons and others has $5d^{3+}$ valence electrons [1, 2]. **e** STM image ($V_s = 5$ mV, $I_t = 3$ nA) of the Pb atoms on the stripe phase. The Pb atoms selectively adsorb on the dimerized Ir site of the stripe phase. The atomic position of stripe phase in **b** and the structural image in **d** were taken from Ref.[1] and [2], respectively.

Replies to the comments of the second referee

The article of Park et al. reports an interesting study of Pb clusters assembled on the charge density wave material IrTe₂.

I read the manuscript with great interest. I think it is a very solid study comparing high quality STM/STS data with state-of-the-art ab-initio calculation. The authors show the importance of the relativistic spin-orbit interaction in explaining the electronic properties of the clusters.

I have not any comment on the scientific aspects of the manuscript. It is clearly written, and I would not have any objection in publishing it in its current state.

However, I find it suitable for a more specialized journal. I do not see sufficient potential for impact and novelty as expected by Nature Communication. I would be happy to change my opinion if these clusters could be controllably manipulated by the STM tip, to achieve a real tailored-made fabrication approach. At present, the clusters are randomly obtained by dosing Pb on IrTe₂. Alternatively, another interesting aspect would be the creation of a well-defined artificial Pb superlattice.

I also find the title a little bit confusing. The article highlights the importance of the spin-orbit coupling. Since Pb is an heavy element, I do not find this aspect very surprising. However, I think that naming these clusters “Dirac molecules” can be rather misleading. It recalls a concept generally used in other contexts (Dirac bands in graphene or topological materials).

To summarize: I think this is a very comprehensive study, but I do not think it fits within the scope of the journal. However, if the editor or the other referees disagree on this aspect, I have no objection in publishing the manuscript in its present state.

We appreciate the positive evaluation of the referee. We agree that the title can confuse the condensed matter physics readers and the “Dirac molecule” is not widely used. In principle, this is a right terminology since the Dirac band means the band solution of the Dirac equation and the Dirac molecule means the molecular orbital solution of the same Dirac equation. The title but can be changed to “relativistic molecules” without hurting the main message. Note that the relativistic effect of Pb molecules is not surprising but such molecules have not been realized so far.

Replies to the comments of the third referee

The authors reported a comprehensive study on the interactions among individual Pb adatoms on IrTe₂ using STM and DFT. The authors concluded that small clusters of Pb in several configurations form molecular orbitals supported by STS spectra, real space STS mapping, and DFT calculations. The results are interesting and likely correct except that the energies of the simulated charge density maps which are shown to resemble experiments do not quite agree with experiments. Still, this study provides a simple platform for exploring artificial states of matter via bottom-up methods. Compared to another bottom-up system of "molecular graphene" from manipulated CO molecules (Nature 483, 306–310, 2012), the drawback in the current work is clearly the lack of flexibility in the inter-atomic distances and lattice structures. For example, could the inter-Pb distance be manipulated by STM tip in this case? I also have the following questions the authors should address before this manuscript is considered for publication.

We appreciate the positive evaluation of the referee and his/her helpful comments. Considering the comment of the referee, we briefly mentioned the limitations of the current approach in comparison to the atom/molecule manipulation. In fact, the most important aspect of the present approach is not its technical merit over the previous works but the uniqueness of the atomic level interaction in this system, which leads to the formation of the unprecedented relativistic molecular structures. This point was reflected in the revised manuscript by carefully rewriting the introduction part. Our detailed replies to the referee's technical questions are given below.

1. The authors claimed, "The direct observations of relativistic molecular orbitals are exceptional". While the DFT calculations provided an explanation for the experimental data, in the strictest sense, the observation should be stated as consistent with relativistic molecular orbitals.

We partly agree on the reviewer's comment in its strictest sense. The direct observation is in its strict sense the direct real space observation (we added "real space" in the corresponding text). However, we note that some molecular orbital shapes (for example, Fig. 2i and those in Fig. 3) cannot be found in the scalar relativistic case. This partly justify the claim of the direct observation.

2. In Fig. 3B, the authors claimed broken A-B sublattice symmetry. I don't see such evidence clearly from the map in Fig. 3B.

As the referee can find below, the broken A-B sublattice symmetry can be found in Fig. 3B and 3D rather easily. Fig. 3D case is very obvious. In Fig. 3B, one can notice the different dI/dV intensity between neighboring Pb atoms of the benzene ring. This is shown more clearly in the figure below, which uses different ways of plotting the same dI/dV intensity data.

3. Does Fig. 3L-N correspond to panels B-D, respectively? Why are Fig. 3L and M different while they are both simulated at 1.28eV as the authors labeled? It looks to me that Fig. 3B-D are taken at very different energies (the black arrows in E), but Fig. 3L-N are given at energies of 1.28 and 1.29 eV, which are close. Though L-N resembles B-D, the large energy difference makes it not convincing.

(Combined with the next comment)

4. Similarly, Fig. S16 B-D seems to have very different energies than what are indicated in Fig. S16 G-I.

We appreciate the referee for pointing this issue out. We agree that the energies of some molecular levels in simulation deviate from the experiment. While the simulation unambiguously confirms the relativistic bonding picture for the smaller molecules, the energy splitting itself is underestimated, by about 0.05 eV for the shortest dimer (Figs. 2j and 2v). This level of discrepancy becomes more important in the case of larger molecules shown in Fig. 3, where, for example, the six molecular levels are expected in a narrow energy window of 0.13 eV for the $|1/2\rangle$ state. That is, we admit that the current simulation is not accurate enough to reproduce the small energy splittings of large molecules. However, the energy hierarchy of the molecular levels are rather well reproduced even in those cases and it is unambiguous that the scalar relativistic calculations give qualitative difference from the experiment. Note also that the STS experiment is thought to pick favorably up the p_z orbital character, which would be the reason to enhance the three orbitals among six for the $|1/2\rangle$ state of the benzene-like molecules. We also find it fairly confusing to provide Fig. 3s of a different energy than Fig. 3i to show the molecular orbital with the edge mode. We revised this part of the figure for a more direct and consistent comparison.

STS spectra and PDOS of the Pb hexamer

Comparison between with and without SOC

5. Explanations are needed for Fig. 3O, T as they are not mentioned in the main texts.

We appreciate the referee for pointing out our own mistake. Following the referee's advice, we added the explanation for the Fig. 3O and T in corresponding caption and refereed in the main text.

6. In the caption of Fig. S13, the authors mentioned spin configurations in Fig.4, while there is no Fig. 4 in this manuscript. The authors also mentioned "unusual spin structure" in the Summary section, which is not discussed at all in the manuscript.

We appreciate the referee for pointing out our own mistakes. The typo of Fig. 4 was corrected to Fig. 3. Various and complex spin configurations arise from the SOC and the substrate interaction in each molecular state of the Pb hexamer. We found the parallel, antiparallel, **rotational** and other complex spin orderings shown partly below. Following the referee's comment, we add a sentence to discuss the spin configuration of the Pb hexamer and provide a supplementary figure (SFig. 13d) for the parallel, antiparallel, and helical spin configurations in the $|1/2, \pm 1/2\rangle$ states. These unusual spin configurations would give more rich physics.

Spin configurations: **Rotational**, antiparallel, and parallel spin configurations in the $|1/2, \pm 1/2\rangle$ states of the Pb hexamer. **a** Charge density, **b** spin configuration, **c** plane projected spin configuration. The numbers denote its energy level.

7. More explanations of Fig. S13 are needed. For example, what are the energies of the simulated charge density distributions? Does charge distributions with SOC reproduce experimental observations better than without SOC?

We appreciate this helpful comment. Following the referee's comment, we modified the Supplementary Fig. 13 and its caption. To describe the molecular states, we used the energy level at Gamma point and the charge densities were plotted also at each all energy at Gamma point. Thus, the hexamer (6 Pb cluster) has 18 states without spin degree of freedom and 36 states for the relativistic case in Supplementary Fig. 13 B and C, respectively. The charge distribution cannot explain the experimental observations such as the broken AB sublattice symmetry and a Kekule-like distortion, and also all STS spectra and dI/dV maps for other Pb clusters are described consistently and clearly with SOC.

Supplementary Fig. 13

FIG. 13: Energy level and charge characters of the benzene-like Pb molecule on IrTe₂-(7×7) with/without spin-orbit coupling. The Fermi level sets to zero. **a** The Γ point energy level for the 6p states of the Pb atom. **b** Charge characters without spin-orbit coupling. **c** with spin-orbit coupling. For the case of spin-orbit coupling, there are two energetically degenerated states (red and blue) but distinguishable in their spin configuration. The number denote the corresponding energy level. **d** Selected spin configurations of helical, antiparallel and parallel in the $|1/2\rangle$ states.

8. In terms of materials systems, the authors are encouraged to provide outlooks on what other adsorbates and substrates are interesting beyond what's demonstrated here.

We appreciate this constructive suggestion. In the theoretical aspect, we already discussed the relativistic effect and interaction strength with the substrate by using other elements of Sn and Tl. However, we can further suggest that the adsorption of magnetic atoms on the present substrate to realize unique magnetic molecules and using the substrates with novel superstructures to work as the templates such as the domain wall networks of 1T-TaS₂, 2H-TiSe₂, MoSe₂, and most interestingly twisted bilayer graphene. This suggestion was added at the end of the summary section.

Summary of changes

(1) We changed the title to avoid confusion.

=> Artificial Relativistic Molecules

(2) We revised sentences in introduction to tone down our claim.

=> The direct atom-by-atom manipulation and the self-assembly of atoms or molecules are two major approaches to realize such atomic scale chains, clusters, and finite lattices. Both methods, however, have their own limitations under given interatomic interactions and fabricating energetically and kinetically unfavorable structures or assemblies has been a huge challenge [13].

In case of the self assembly of supported clusters, such limitations may be overcome by templates, which provide unusual growth environments to produce otherwise unfavored molecular structures.

(3) We revised a sentence in result to provide detail explanation of the Pb-substrate interaction.

=> Our calculation shows that a Pb adatom is partly ionized by donating 6p electrons into the substrate to shift the 6p states to unoccupied states (Fig. 2o and Fig. 2w). As shown in Fig. 2w, the relativistic effect of SOC split the 6p states into \$6p_{1/2}\$ and \$6p_{3/2}\$ Dirac orbitals and the latter splits further due to the substrate-induced splitting of in-plane (\$p_{xy}\$ ) and out-of-plane (\$p_z\$ ) orbitals. Consequently, three main peaks correspond to the relativistic p orbitals of $p_{1/2}$ ($j=1/2, m_j = \pm 1/2$) and $p_{3/2}$ ($j=3/2, m_j = \pm 3/2, \pm 1/2$), respectively.

=> This is clearly shown in the experiment (Figs. 2h and 2i) and in the calculation (Figs. 2t and 2u). Further details of the interaction with the substrate will be discussed below, which become important for the understanding of the relativistic effect on the adsorption energy.

=> The $2a_0$ distance is optimized by two competing energy contributions, the ionic (or dipole) repulsion and the orbital overlap energy gain of adatom-substrate hybridized states. The comparison of the calculated differential charges with and without SOC reveals two sophisticated calculated differential charges reveal two SOC effects for those contributions;

(4) We added a sentence in result to compare the experiment and theory.

=> Our DFT calculation predicts (Figs. 2r and 2V) the lowest molecular orbital splittings of 0.05 and 0.18 eV for the longer and shorter dimers, respectively, in reasonable agreement with the experiment. The present DFT calculation seems to underestimate the molecular bonding interaction by about 0.05 eV. These energy scale is.....

(5) We revised sentences in result to deliver more clearly the simple bonding picture of the Pb dimers.

=> The molecular levels of a Pb dimer are well explained by the simple tight-binding interaction of Dirac atomic orbitals [27] perturbed weakly by the substrate as discussed above. That is, three relativistic p orbitals split into the combination of molecular orbitals in gerade (g) and ungerade (u) symmetries (see Fig. 2w).

(6) We added sentences in result to discuss the energies of some molecular levels in simulation deviate from the experiment.

=> Our calculation reproduces well the spatial characteristics of these molecular orbitals, which are indeed due to the relativistic character and the interaction with the substrate (Fig. 3l-n and Supplementary SFig.13). Note, however, the remaining energetic discrepancy between experiment and calculation. As mentioned above the present calculations underestimate the molecular bonding interaction and the calculated spectral features are almost degenerates at around 1.3 eV for the benzene-like molecule. We think that the difference is at least partly due to the limitation of our model in taking into account of the substrate electronic states in its charge-ordered correlated state [26]. Another interesting molecule is the Pb pentamer (Fig. 3p).

(7) We revised a sentence and the caption of Fig. 3 in result to explain and discuss the theoretical DOS.

=> Another interesting molecule is the Pb pentamer (Figs. 3f-3j), whose Dirac molecular orbitals are also reasonably well simulated in the present calculation (Figs. 3p-3t).

(8) We added a sentence in result to discuss the spin configurations of the Pb hexamer.

=> This molecule can be compared with a rare example of a planar B₇ molecule [38] except for a strong SOC. While not accessibly by the present experiments, the benzene-type Dirac molecular orbitals fabricated here have unique and interesting spin configurations (see Supplementary SFig. 13), which may be exploited further. It becomes very obvious that the present artificial Pb molecules introduce unprecedented molecular configurations combined with the strong SOC.

(9) We added sentences in summary to provide outlooks on what other adsorbates and

substrates are interesting.

=> Beyond the relativistic chemistry, the unusual bonding and spin structure of these relativistic molecular orbitals may be combined with the novel electronic properties of the substrate such as the charge ordering and the emerging superconductivity [26] to lead to a new type of a quantum system. In addition to the versatile technique of atom-by-atom manipulation, where the interatomic distance may be varied, the use of various types of 2D superstructures as templates would definitely extend the potential of molecular or cluster self assemblies. Interesting 2D superstructures can include twisted bilayer graphene (ref) and domain-wall or twin-boundary networks of transition metal dichalcogenides (ref). The templated self assembly of magnetic atoms on such substrates can be very interesting due to the proximity to the novel 2D electronic states of substrates.

(10) We revised the Fig. 3i and 3t to give more direct and consistent comparison between experiment and theory.

(11) We added three references.

- E. J. Mele, Phys. Rev. B **81**, 161405(R) (2010).
- D. Cho, et. al., Nat. Commun. **7**, 10453 (2016).
- Y. Ma, et. al. ACS Nano **11**, 5130 (2017).

(12) Minor changes.

- In case of a dimer with a shorter Pb-Pb distance, an energy splittings (0.23 eV) of molecular bonding and antibonding states is observed clearly for the $6p_{1/2}$ (Fig. 2j).
- The resulting Dirac molecular orbitals are clearly differentiated from their scalar-relativistic counterparts of σ and π molecular orbitals.
- The direct and real space observation of relativistic molecular orbitals are exceptional,
- Although a Pb dimer has a much longer bond length (7 Å) than other relativistic molecules [33-35] and there is a substantial adatom-substrate interaction,...
- It is important to point out that the attractive interaction between Pb adatoms is only due to SOC and that the $2a_0$ distance imposed by the substrate is optimal for the formation of a Pb dimer (Fig. 2x) in our calculations.
- The Pb₅ molecule features the edge states at both truncated ends between the bonding and antibonding levels of three Dirac p orbitals (see Supplementary SFig. 14)
- The formation of a filled-benzene-ring molecule is also highly unusual (Fig. 1j), where the interatomic overlap is much enhanced by the central Pb atom with unique bonding configurations (Supplementary SFigs.15-17).

Change in Supplementary Materials

- (1) We revised the Supplementary Fig. 1 and its caption.
- (2) We revised the Supplementary Fig. 13 and its caption.
- (3) We added the Supplementary Fig. 14.

SFIG. 14: Energy level and charge characters of the Pb pentamer on IrTe_2 - (7×7) . **a** The Γ point energy level for the $6p$ states of the Pb atom. **b** Projected density of states. **c** Charge characters. The arrows indicate the edge states of three relativistic orbitals. The Fermi level sets to zero.

REVIEWERS' COMMENTS:

Reviewer #1 (Remarks to the Author):

The authors have answered most of my concerns in their reply. I still have the following comments – after addressing these concerns, I would be happy to recommend publication in Nature Communications.

1. In their reply, the authors discuss the happy accident that Pb 6p levels are within the band gap of the IrTe₂ substrate. They should show some larger bias range spectra including spectra on the IrTe₂ substrate, so that we can verify that this is a fact also experimentally.

2. The authors attribute the difference between the calculated and experimental LDOS maps on the Pb dimer to “noise” (in the dI/dV signal). I don’t agree with this; first of all, one should be able to relatively easily obtain dI/dV maps with very high signal to noise ratio with a well-working low-temperature STM. Secondly, noise (unless the raw data is heavily smoothed – is it?) should not cause distortions in the images. Consequently, I don’t think one can conclude “This is clearly shown in the experiment (Figs. 2h and 2i) and in the calculation (Figs. 2t and 2u).”

Reviewer #3 (Remarks to the Author):

The authors have performed a mostly satisfying revision of the manuscript based on my first round of comments, even though the theory parts still lack quantitative agreements with experiments. Realizing the limitations of current DFT simulation tools and the interesting results reported in this manuscript, I would recommend publication in the current form.

Reviewer #1 (Remarks to the Author):

Review of Condon et al. (resubmitted)

Recommendation: Reject

General comments:

The authors have still not taken my comments into account. They recognise in their answer that their "pseudo-warming" experiment is not realistic and very different from actual climate projections, but the article has not been adapted. For instance, the manuscript includes the following sentences describing the set-up of the experiments:

"The three warming scenarios selected here span the range of projections for the CONUS over the 21st century; 1.5°C is the expected warming by the middle of the 21st century if current warming trends continue, 2 and 4°C further provide a range of possible warming scenarios by the end of the 21st century." This is too little context. The authors could for instance have provided actual analyses of climate projections for comparison with their "pseudo-warming experiments".

Response:

We are sorry to hear that the reviewer felt that this sentence indicated that we had not listened to their original review. We disagree with this assessment though given that we (1) added significant discussion to the original submission regarding the purpose of using our 'pseudo-warming approach at Lines 99-115 and (2) did provide references and figures based on actual analyses of IPCC projections to support this choice in our last response.

As we discussed extensively in our previous response, we chose these warming levels as representative perturbations. In our previous response we provided figures and references indicating how we decided that these temperature perturbations would be representative. More importantly we stand by our approach as a means to illuminate the fundamental interaction of groundwater processes with systematic warming and therefore, provide a critical advancement in understanding how surface water and groundwater interactions will change due to warming. Incorporating transient and heterogeneous projections directly from IPCC model outputs would not yield a better understanding of the groundwater response to warming and would in fact limit our ability to isolate this critical connection. Nevertheless, we have made every effort to be transparent about this approach and the fact that we are not making projections here but evaluating sensitivity.

Finally, while the reviewer pulled out one sentence of our description in their comment above, we would like to point out that in response to their previous comments we did expand on the description of our methodology as follows.

We compare a historical baseline climate scenario with three perturbed scenarios with uniform warming of 1.5, 2, and 4°C. The purpose of the uniform so-called, pseudo-warming (Rasmussen et al., 2011) perturbations applied here is to directly evaluate the sensitivity of the terrestrial hydrologic system to varying degrees of warming in a series of controlled experiments. We acknowledge that climate projections show spatially variable warming trends, and temporal variability in warming especially with extreme events (Hayhoe et al., 2018), our goal is not to capture this variability but to evaluate response to a range of long-term warming possibilities.

The three warming scenarios selected here span the range of projections for the CONUS over the 21st century: 1.5°C is the expected warming by the middle of the 21st century if current warming trends continue (IPCC, 2018), 2 and 4°C further provide a range of possible warming scenarios by the end of the 21st century (Hayhoe et al., 2018). Some parts of the country are projected to warm faster or slower than these national averages. For example, the Northeast is projected to warm faster than the Southwest (Hayhoe et al., 2018; Vose et al., 2017). Therefore, our 4°C scenario may be a shorter-term likelihood for some parts of the country than others. The three temperatures chosen here are not intended to be national projections for a specific time period. We seek to quantify the response to warming separate from considerations of variability and uncertainty in warming projections therefore these values were chosen to represent a range of warming that is reasonable across CONUS.

We think this description is transparent about the limitations and assumptions that the reviewer brought in their previous review. We thank them for those comments which we do feel improved the clarity of the manuscript.

Another issue (which I had not noted in my first review) is that it appears that the authors do not consider possible direct effects of CO₂ on evapotranspiration. Because enhanced CO₂ could increase water-use efficiency of plants, this can reduce evapotranspiration under a warmer climate (e.g. Lemordant et al. 2016).

Response:

The reviewer brings up an important point that echoes Reviewer 4's original comment: "You should mention factors that may counter your assumptions. There are studies that suggest plant stomas tend to close, reducing ET, as CO₂ levels increase." Based on that comment from Reviewer 4 we added the following text to the original revised manuscript (Line 312):

"Dynamic groundwater interactions will likely affect ecosystem productivity (Burkett et al., 2005) or conversely be affected by warming induced changes to hydraulic conductivity (Constantz, 1982; Reitz & Sanford, 2019), both of which were not considered and may counteract some of the sensitivity found here."

We would like to point out that Reviewer 4 accepted this change and strongly endorses the revised manuscript. Still, we acknowledge the point that and agree that this point could be further highlighted. We have further revised this as follows:

Dynamic groundwater interactions will likely affect ecosystem productivity (Burkett et al., 2005) or conversely be affected by warming induced changes to hydraulic conductivity (Constantz, 1982; Reitz & Sanford, 2019). Also, CO₂ enrichment may increase water use efficiency and partly compensate for increased evaporative demand caused by warming (Guerrieri et al., 2019; Lemordant et al., 2016). These interactions were not considered in this study and may counteract some of the sensitivity found here.

Finally, we would like to stress that although this is an important consideration, the future impacts of CO₂ enrichment are still a very active research area with much uncertainty. CO₂ enrichment can have competing impacts on total plant water as increased leaf area index may counter increased water use efficiency. Furthermore, as pointed out in a recent PNAS paper: "Multiple lines of evidence suggest that plant water-use efficiency (WUE)—the ratio of carbon assimilation to water loss—has increased in recent decades. Although rising atmospheric CO₂ has been proposed as the principal cause, the underlying

physiological mechanisms are still being debated, and implications for the global water cycle remain uncertain” (Guerrieri et al., 2019). Modeling how these small scale processes can be accurately captured at the scales we are modeling remains an active area of research and model development (Fatichi et al., 2016; Lin et al., 2015; Xu et al., 2013). Thus, while we agree that these interactions are worth mentioning we stand by our decision not to incorporate this into our controlled groundwater sensitivity analysis.

The study seems too conceptual and far from a realistic set-up. It is based on a single model, while it is well known that different hydrological models can present very different responses to climate forcing (e.g. Prudhomme et al. 2014). It also does not include changes in radiation, atmospheric humidity or precipitation, which would all be modified under realistic climate-change projections.

Response:

The referred Prudhomme et al. paper compares hydrologic projections from several global impact models (GIMs) and explicitly demonstrates how different hydrologic representations produce large outcome uncertainty based on model structure, i.e. process representation (here we should note that none of the models in the Prudhomme paper include dynamic groundwater lateral flow and interaction with surface water, the specific process focus of our manuscript). The large discrepancies between models shown by this paper precisely demonstrates why we need to improve our processed based understanding to provide better constrained and more actionable hydrologic projections.

Our work isolates, and therefore quantifiably demonstrates, the importance of dynamic groundwater interactions in watershed response to warming. Representing, and more importantly understanding, the role of storage changes in a changing climate is a consistent limitation in existing global models and critical need that has been expressed repeatedly in the literature including in high profile journals (B.E. Jiménez Cisneros et al., 2014; Humphrey et al., 2018; Jung et al., 2017; Lemordant et al., 2016; Ove Hoegh-Guldberg et al., 2018). We base our work on a single model because ours is the only model that explicitly representing these groundwater interactions across the US. We have added the following text to the revised manuscript to highlight this point more clearly and point to the potential for this type of comparison with future work:

While currently only one, integrated hydrologic model exists for North America, as more simulations of this type are undertaken a multi-model approach(Prudhomme et al., 2014) could be used to further study the impact of conceptual model uncertainty on response or to propagate warming projections to groundwater.

While we have designed a controlled numerical experiment, this approach is far from conceptual. We are simulating 3D variably saturated flow across millions of square kilometers. This is the most computationally advanced hydrologic simulation currently available for the US. We are choosing a simple perturbation to control for the number of degrees of freedom in our experiment and provide tractable results that can be useful to other models that either neglect subsurface flow or rely on more conceptual groundwater representations. This type of controlled numerical experiment is a very common way that the research community is working through complex interactions. As just one example, the Lemordant et al. (2016) paper the reviewer cited above takes the same approach; holding other variables constant while systemically changing CO₂ levels to, “to estimate the sole impact of CO₂ physiological effects” (Lemordant et al., 2016).

We recognize the reviewer's fundamental disagreement of our work, in that by not representing all possible processes that contribute hydrological conditions we are unable to produce a valid hydrologic projection as a result of climate change. However, what the reviewer has chosen to neglect is that process understanding is foundational to scientific knowledge and work such as ours that aims to understand the underlying processes are specifically relevant and of particular interest to the climate community for the reasons pointed out by the Prudhomme et al., 2014 paper.

Based on these various points, I cannot recommend publication of this manuscript in Nature Communications. The article would be suitable for a disciplinary journal (e.g. ERL, GRL). But this study is not suitable at present for a high-impact journal, because it is not clear enough to which extent the results would be robust if a different experimental set-up or hydrological model were used.

Response: We are sorry that the reviewer does not appreciate the contributions of this work and the significant and unprecedented nature of the simulations we developed here. As noted above, there are no other hydrologic models simulating the integrated processes we consider over the Continental US. We are using a rigorous and computationally intensive approach to quantify interactions based on our best scientific understanding of subsurface flow and groundwater surface water interactions. This is valuable to a broad community precisely because other models are not incorporating these processes and our global community is struggling to understand what the role of storage will be in a warmer climate.

References:

- Lemordant, L., et al. 2016: Modification of land-atmosphere interactions by CO2 effects: Implications for summer dryness and heat wave amplitude. *Geophys. Res. Lett.*, 43, doi:10.1002/2016GL069896.

- Prudhomme, C., et al. 2014: Hydrological droughts in the 21st century, hotspots and uncertainties from a global multimodel ensemble experiment. *PNAS*, www.pnas.org/cgi/doi/10.1073/pnas.1222473110

Reviewer #3 (Remarks to the Author):

Many thanks for addressing in detail all my comments (reviewer #3). I'm happy with all your proposed changes.

John Bloomfield

Response:

Thank, you for your suggestions John. We think they improved the revised manuscript and we are happy that you agree.

Reviewer #4 (Remarks to the Author):

The authors have made a conscientious effort to respond to the three reviewers comments on their manuscript. I agree with the authors concerning the first reviewer's comments, which were the most critical of the three. The reviewer argues that the authors make a grave error by treating all of the conterminous US as having increasing temperatures that are constant in space, not taking into account N-S or E-W trends across the continent, which might be substantial, and it is implied this is a misunderstanding of the authors and limits the usefulness of the results. But my thought before even reading the authors' response was that they knew this to be the case but were not trying to simulate specific scenarios projected by the climate models, but were trying to look at a more general response of the hydrology to increasing temperatures. When taken in this context their numerical experiments give meaningful results. And indeed the author's response indicated that was their intent to show all along.

The second and third reviewer's were less critical, but offered advice for more meaningful discussions of the results, and the author's did a good job of incorporating these suggestions into the text of the manuscript--including additional tables to illustrate changes in ET as a percent of recharge.

I believe that given the changes made by the authors that this manuscript should now be accepted by the journal for publication.

Response:

Thank, you for the thoughtful consideration of our work. We agree with your assessment and have made every effort to be transparent about our methodology and assumptions at every step of the way. We appreciated all of the suggestions we received in the first round and we think they improved the manuscript significantly.

References

- B.E. Jiménez Cisneros, Oki, T., Arnell, N. W., Benito, G., Cogley, J. G., Petra, D., et al. (2014). *Freshwater resources. In: Climate Change 2014: Impacts, Adaptation, and Vulnerability. Part A: Global and Sectoral Aspects. Contribution of Working Group II to the Fifth Assessment Report of the Intergovernmental Panel on Climate Change*. Retrieved from Cambridge, United Kingdom and New York, NY USA:
- Burkett, V. R., Wilcox, D. A., Stottlemeyer, R., Barrow, W., Fagre, D., Baron, J., et al. (2005). Nonlinear dynamics in ecosystem response to climatic change: Case studies and policy implications. *Ecological Complexity*, 2(4), 357-394.
<http://www.sciencedirect.com/science/article/pii/S1476945X05000334>
- Constantz, J. (1982). Temperature Dependence of Unsaturated Hydraulic Conductivity of Two Soils 1. *Soil Science Society of America Journal*, 46(3), 466-470.
- Fatichi, S., Pappas, C., & Ivanov, V. Y. (2016). Modeling plant-water interactions: an ecohydrological overview from the cell to the global scale. *Wiley Interdisciplinary Reviews: Water*, 3(3), 327-368. <http://dx.doi.org/10.1002/wat2.1125>
- Guerrieri, R., Belmecheri, S., Ollinger, S. V., Asbjornsen, H., Jennings, K., Xiao, J., et al. (2019). Disentangling the role of photosynthesis and stomatal conductance on rising forest water-use efficiency. *Proceedings of the National Academy of Sciences*, 116(34), 16909-16914. <http://dx.doi.org/10.1073/pnas.1905912116>
- Hayhoe, K., Wuebbles, D. J., Easterling, D. R., Fahey, D. W., Doherty, S., Kossin, J., et al. (2018). *Our Changing Climate. In Impacts, Risks, and Adaptation in the United States: Fourth National Climate Assessment, Volume II* Retrieved from Washington, DC, USA:
- Humphrey, V., Zscheischler, J., Ciais, P., Gudmundsson, L., Sitch, S., & Seneviratne, S. I. (2018). Sensitivity of atmospheric CO₂ growth rate to observed changes in terrestrial water storage. *Nature*, 560(7720), 628-631. <http://dx.doi.org/10.1038/s41586-018-0424-4>
- IPCC. (2018). *Summary for Policymakers. In: Global warming of 1.5°C. An IPCC Special Report on the impacts of global warming of 1.5°C above pre-industrial levels and related global greenhouse gas emission pathways, in the context of strengthening the global response to the threat of climate change, sustainable development, and efforts to eradicate poverty*. Retrieved from Geneva, Switzerland:
- Jung, M., Reichstein, M., Schwalm, C. R., Huntingford, C., Sitch, S., Ahlström, A., et al. (2017). Compensatory water effects link yearly global land CO₂ sink changes to temperature. *Nature*, 541(7638), 516. <http://dx.doi.org/10.1038/nature20780>
- Lemordant, L., Gentine, P., Stéfanon, M., Drobinski, P., & Fatichi, S. (2016). Modification of land-atmosphere interactions by CO₂ effects: Implications for summer dryness and heat wave amplitude. *Geophysical Research Letters*, 43(19).
<http://dx.doi.org/10.1002/2016GL069896>
- Lin, Y.-S., Medlyn, B. E., Duursma, R. A., Prentice, C. I., Wang, H., Baig, S., et al. (2015). Optimal stomatal behaviour around the world. *Nature Climate Change*, 5(5), 459-464.
<http://dx.doi.org/10.1038/nclimate2550>

- Ove Hoegh-Guldberg, Jacob, D., Taylor, M., Bindi, M., Brown, S., Camilloni, I., et al. (2018). IPCC SR1.5: Chapter 3: Impacts of 1.5°C global warming on natural and human systems.
- Prudhomme, C., Giuntoli, I., Robinson, E. L., Clark, D. B., Arnell, N. W., Dankers, R., et al. (2014). Hydrological droughts in the 21st century, hotspots and uncertainties from a global multimodel ensemble experiment. *Proceedings of the National Academy of Sciences*, *111*(9), 3262-3267. <http://dx.doi.org/10.1073/pnas.1222473110>
- Rasmussen, R., Liu, C., Ikeda, K., Gochis, D., Yates, D., Chen, F., et al. (2011). High-Resolution Coupled Climate Runoff Simulations of Seasonal Snowfall over Colorado: A Process Study of Current and Warmer Climate. *24*(12), 3015-3048. <https://journals.ametsoc.org/doi/abs/10.1175/2010JCLI3985.1>
- Reitz, M., & Sanford, W. E. (2019). Estimating quick-flow runoff at the monthly timescale for the conterminous United States. *Journal of Hydrology*, *573*, 841-854. <http://pubs.er.usgs.gov/publication/70203038>
- Vose, R. S., Easterling, D. R., Kunkel, K. E., LeGrande, A. N., & Wehner, M. F. (2017). Temperature changes in the United States. In D. J. Wuebbles, D. W. Fahey, K. A. Hibbard, D. J. Dokken, B. C. Stewart, & T. K. Maycock (Eds.), *Climate Science Special Report: Fourth National Climate Assessment, Volume I* (pp. 185-206). Washington, D.C.: U.S. Global Change Research Program.
- Xu, C., McDowell, N. G., Sevanto, S., & Phytologist, F.-R. A. (2013). Our limited ability to predict vegetation dynamics under water stress. *New Phytologist*. [http://dx.doi.org/10.1111/nph.12450@10.1002/\(ISSN\)1469-8137\(CAT\)FeatureIssues\(VI\)Droughtinducedforestmortality](http://dx.doi.org/10.1111/nph.12450@10.1002/(ISSN)1469-8137(CAT)FeatureIssues(VI)Droughtinducedforestmortality)

Replies to the comments of the first referee

The authors have answered most of my concerns in their reply. I still have the following comments – after addressing these concerns, I would be happy to recommend publication in Nature Communications.

1. In their reply, the authors discuss the happy accident that Pb 6p levels are within the band gap of the IrTe₂ substrate. They should show some larger bias range spectra including spectra on the IrTe₂ substrate, so that we can verify that this is a fact also experimentally.

We appreciate the reviewer's suggestion and we agree that it is required to fairly compare between experiment and theory. The larger bias range spectra were already provided in the Supplementary Figure 5. However, we extended this figure for a more direct comparison with theory as shown also above.

As shown here, the Pb 6p state is located on the flat region of the spectrum for the IrTe₂ substrate (black line in experiments), which corresponds well to the gap (2D single layer limit) or the low DOS region (3D bulk) of the IrTe₂ substrate in the calculation. The overall difference between the STS spectra and theoretical DOS comes mostly from the tunneling background, which is normally parabolic from the Fermi energy.

2. The authors attribute the difference between the calculated and experimental LDOS maps on the Pb dimer to "noise" (in the dI/dV signal). I don't agree with this; first of all, one should be able to relatively easily obtain dI/dV maps with very high signal to noise ratio with a well-working low-temperature STM. Secondly, noise (unless the raw data is heavily smoothed – is it?) should not cause distortions in the images. Consequently, I don't think one can conclude "This is clearly shown in the experiment (Figs. 2h and 2i) and in the calculation (Figs. 2t and 2u)."

We appreciate the referee for pointing out our insufficient explanation. Reflecting the concern of the referee (and considering all the discussion given below), we can tone down the sentence indicated by the referee, "This is clearly shown in the experiment (Figs. 2h and 2i) and in the calculation (Figs. 2t and 2u)" into "This is reasonably confirmed in the experiment (Figs. 2h and 2i) and in the calculation (Figs. 2t and 2u)." Particular for the loss of the mirror symmetry perpendicular to the dimer axis, which is noted by the referee, is unambiguous in the above images but only the other mirror symmetry about the dimer axis seems rather unclear in Fig. 2i.

As we stated in our previous reply, we want to emphasize again that there is no atomic distortion of Pb molecules including Pb dimers and hexamer as identified by the topographic images (see Supplementary Figure. 2). The "noise" or "disorder" in STS spectral images, thus, has electronic origin. Those are rather natural in dI/dV measurements since STS is a differential measurement. That is, dI/dV measurements always carry much higher noise than the integrated measurements of the STM topographies. This is especially true for a coarse graining spatial sampling as we did in the present measurements (see the pixel size in the corresponding figures as also shown below). The degree of irregularity ("noise") in the dI/dV maps can be easily viewed in these standard data for an isolated adatom, which is much larger than that in topography (A).

Supplementary Figure 3

However, to be more careful on this issue, we note that there can be more systematic deviation than statistical irregularity or disorder. If one sees the difference between Fig. 2h and 2i in the supplements, then the off-axis deviation of the STS map from the calculated LDOS seems more systematic in Fig. 2i than in Fig. 2h.

Fig. 2h and 2i. Purple dashed (strong) and solid (weak) lines are guides for eye.

This difference may be due to the facts that (i) there are many degenerate levels are involved in a narrow energy range in the calculation (see the figure below, right column) and (ii) the substrate electronic structure, especially the charge order, is not fully accounted for in the present calculation. That is, there are limitations in both experiment (energy resolution to resolve nearly degenerate energy levels) and theory (capturing the electronic states of the substrate accurately). These points are already mentioned in our manuscript. However, we stress again that the shape irregularity in STS maps and the possible systematic deviation are still in a manageable level to draw major qualitative conclusions through the comparison between the experiments and the calculation.

$\sqrt{3}a_0$ Pb dimer

Spin summed

Spin decoupled

spin component 1

spin component 2

Summary of changes

(1) We revised a sentence in result to tone down our claim.

=> This is **reasonably confirmed** in the experiment (Figs. 2h and 2i) and in the calculation (Figs. 2t and 2u).